# Evaluation of Resin Infiltration, Fluoride and the Biomimetic Mineralization of CPP-ACP in Protecting Enamel after Orthodontic Inter-Proximal Enamel Reduction

**DOI:** 10.3390/biomimetics8010082

**Published:** 2023-02-14

**Authors:** Naser Almansouri, Ahmed Samir Bakry, Mona Aly Abbassy, Amal Ibrahim Linjawi, Ali Habib Hassan

**Affiliations:** 1Department of Orthodontics, Faculty of Dentistry, King Abdulaziz University, Jeddah 21589, Saudi Arabia; 2Restorative Dentistry Department, Faculty of Dentistry, King Abdulaziz University, Jeddah 21589, Saudi Arabia

**Keywords:** interproximal reduction, trans microradiography, enamel, caries, orthodontics

## Abstract

Background: This study investigated the effect of using different agents for protecting enamel proximal surfaces against acidic attack after interproximal reduction (IPR) using the trans micro radiography technique. Methods: Seventy-five sound-proximal surfaces were obtained from extracted premolars for orthodontic reasons. All teeth were measured miso-distally and mounted before being stripped. The proximal surfaces of all teeth were hand stripped with single-sided diamond strips (OrthoTechnology, West Columbia, SC, USA) followed by polishing via Sof-Lex polishing strips (3M, Maplewood, MN, USA). Three-hundred micrometers of enamel thickness was reduced from each proximal surface. The teeth were randomly divided into 5 groups: group 1 (control un-demineralized) received no treatment, group 2 (control demineralized) had their surfaces demineralized after the IPR procedure, group 3 (fluoride) specimens were treated with fluoride gel (NUPRO, DENTSPLY, Charlotte, NC, USA) after the IPR, group 4 (Icon) resin infiltration material (Icon Proximal Mini Kit, DMG, Bielefeld, Germany) was applied after IPR, group 5 (MI varnish) specimens were treated with Casein phosphopeptide-amorphous calcium phosphate (CPP-ACP) containing varnish (MI Varnish, G.C, USA, St. Alsip, IL, USA) after the IPR. The specimens in (groups 2–5) were stored in a 4.5 pH demineralization solution for 4 days. The trans-micro-radiography (TMR) technique was conducted to evaluate the mineral loss (∆Z) and lesion depth of all specimens after the acid challenge. The obtained results were analyzed statistically using a one-way ANOVA at a significance level of α = 0.05. Results: The MI varnish recorded significant ∆Z and lesion depth values compared to the other groups *p* > 0.05. There was no significant difference in ∆Z and lesion depth between the control demineralized, Icon, and fluoride groups *p* < 0.05. Conclusion: The MI varnish increased the enamel resistance to acidic attack, and thus can be considered an agent capable of protecting the proximal enamel surface after IPR.

## 1. Introduction

Orthodontic treatment is one of the most important treatment modalities utilized to correct craniofacial defects in populations worldwide; however, treating patients with fixed orthodontic appliances leads to an increase in the streptococcus count in saliva multiple folds when compared to healthy individuals [1]. A recent study showed that orthodontic treatment increases the risk of developing caries which is directly correlated to the duration of the orthodontic treatment, reaching 72% after two years of initiating the orthodontic treatment [2]. Previous studies showed that the first signs of enamel demineralization appear 1 month after bonding the fixed orthodontic appliances to the teeth, which is manifested as white spots on the enamel [3]. White spot lesions are subsurface demineralized enamel regions in which enamel minerals have been depleted by the action of acids of the bacterial biofilm [3]. The superficial remineralization exerted by the fluoride on the outer enamel surface preserves the outer enamel surface from being eroded by the action of the bacterial biofilm acids; however, the subsurface enamel remains demineralized. The subsurface demineralization process deprives the enamel rods of lots of their minerals, creating gaps between the enamel rods that affect the refractive index of light falling on that area giving the appearance of white opaque spots at these areas [3]. Recent histological and chemical analyses of white spot demineralization lesions (WSLs) that occurred naturally in an upper molar extracted for orthodontic reasons showed that there were significant changes emerging inside this zone [4]. The mineral density, Young’s modulus, hardness, and average roughness parameters were depleted in the zone of the white demineralization and the area of enamel and dentin bordering it [4]. The researchers compared the obtained results to the control sound areas of the same tooth and confirmed their findings. Moreover, the maps of mechanical properties showed that the enamel outside of the WSL zone was weakened despite its appearance as being sound on optical and bitewing X-ray images [4]. Additionally, the average roughness at the cross-sectional area of the white spot lesion zone was significantly higher when compared to areas of sound enamel. The chemical analysis of these areas using Raman spectroscopy detected changes in the organic and inorganic phases of enamel in the zone of the WSLs [4]. Previous in vitro studies on artificially induced WSLs utilizing the TMR technique showed that these lesions suffer from a depletion in their mineral density and that it forms a definite lesion depth that can be accurately quantified using the TMR technique [5,6,7].

Moreover, previous reports suggested the difficulty of treating such lesions by topical fluoride application; additionally, these lesions will rapidly be transformed into cavitated enamel lesions if the proper oral hygiene measures are not maintained by the patient [3]. Thus, there is great importance in conducting interproximal reduction (IPR) procedures in many of those cases to avoid extraction or expansion and relieve the mild or even moderate crowding, which may lead to shortening the treatment procedures. Moreover, the widespread use of orthodontic clear aligners increased the frequency of utilizing the IPR technique to achieve orthodontic treatment plan goals in the shortest possible time [8].

However, IPR means that the outer protective prismless enamel layer which is approximately 300 µm thick [9] may be disturbed, causing a decrease in its acid resistance and vulnerability to developing caries [10]. The available guidelines in the literature regarding the IPR show large controversies; some guidelines show that this procedure can be done for both proximal surfaces up to 500 µm [11] and that polishing the surface of enamel may be sufficient to protect the enamel from further acidic attack [11], whereas other clinical studies show that the complete removal of proximal enamel is also possible [12]. On the other hand, it was clearly demonstrated that (IPR) procedures conducted with careful polishing of the enamel produce permanent farrows in enamel that were retentive to bacterial biofilm which was difficult to be cleaned by flossing [13].

These controversial recommendations related to IPR procedures make it important to apply protective agents on enamel surfaces after the IPR procedures to avoid the development of caries as a precautionary measure. Proposed agents for this purpose may include the fluoride topically applied agents, Casein phosphopeptide-amorphous calcium phosphate (CPP-ACP) [14,15], and the resin infiltration technique as it may protect the enamel surface by forming a mechanical barrier that may prevent the penetration of the cariogenic acids into the enamel structure [16].

The lack of clinically evident correlation between the IPR procedures and caries prevalence may be attributed to the complexity of the caries process that depends on cariogenic bacterial biofilm, fermentable sugar and susceptible teeth [17], thus it may be proposed that selecting a small clinical sample size [12], having access to fluoridated water [18], or in countries having concerted dental checkups and a recall system [12,18] may lead to eliminating the first and/or second factors necessary for the caries incidence to occur and thus reporting research results that cannot be replicated in other communities lacking the aforementioned facilities [13]. Consequently, there is an absolute necessity to set up new clear guidelines based on accurate experimental methods to conduct the IPR procedures worldwide to avoid the development of caries in the interproximal surfaces during or after concluding the orthodontic treatment phase.

To fairly assess the potential of any protective agent in preventing the possible development of caries lesions after interproximal reduction, an accurate technique, such as transversal microradiography (TMR), should be employed as it is considered to be the golden standard in demineralization/remineralization experiments [6,14,19,20].

The objective of this study was to compare the effectiveness of different agents (fluoride gel, CPP-ACP containing varnish, and resin infiltration) for protecting enamel proximal surfaces against acidic attack after IPR using TMR. The null hypothesis adopted in the current study is that the tested materials will not protect the enamel surface from a simulated cariogenic acidic attack in vitro.

## 2. Materials and Methods

### 2.1. Sample Inclusion and Exclusion Criteria

Seventy-five sound-extracted human premolars were collected after obtaining ethical approval from the Ethical Committee at the Faculty of Dentistry [Ethical no. 055-02-19]. Teeth with caries, restorations, hypoplasia, white spots, or cracks were excluded. All extracted teeth were cleaned, scaled, and checked under a microscope for any white spots or cracks and preserved in 0.1% thymol solution at room temperature. The number of specimens assigned to each group was estimated according to previous literature and according to the 80% power of test using software (G-power version 3.1.9.2) [21]. Each of the five specimens was placed in a vial and coded. The groups were entered in an Excel sheet (Excel 2007; Microsoft, Redmond, WA, USA) and plotted against the code numbers. Using the randomization option in Excel, the codes of the vials were assigned to the experimental groups. Inter-examiner calibrations between an expert in conducting and interpreting the TMR experiments and a member of the research team of the current experiment were done by examining specimens from the pilot study and comparing the observed values of lesion depth and mineral loss for the same specimens by the two researchers using Cohen’s kappa (κ) test operating on SPSS software *p* < 0.05. This was followed by Intra-examiner calibration for the same member of our research team to guarantee the consistency of his/her observation using Cohen’s kappa (κ) test operating on SPSS software *p* < 0.05.

### 2.2. Sample Preparation and Groups

Each tooth had its maximum mesiodistal diameter measured using a digital caliper. The teeth were mounted in a compound wax on plaster blocks. All proximal enamel surfaces (n = 75 surfaces) were hand stripped with a diamond-coated metal strip (MiniStripper, ortho technology, Tampa, FL, USA). The enamel thickness removed from each proximal surface was approximately 300 µm. These surfaces were polished with polishing strips (Sof-Lex™ Finishing Strips, 3M, Maplewood, MN, USA) according to the manufacturer’s instructions. Galaxy IPR; Gauge Set (OrthoTechnology, Tampa, FL, USA) was used to confirm the amount of enamel reduction.

The proximal enamel surfaces were then randomly divided into groups according to the protective agents applied after stripping. The agents’ composition and modes of application are summarized in Table 1. The proximal enamel surfaces were then randomly divided into 5 groups according to the protective agents applied after the IPR procedure. Each group comprised 15 specimens.

Group 1 (control un-demineralized group): the enamel proximal surfaces were neither reduced nor challenged with the acidic challenge.

Group 2 (control demineralized group): the proximal enamel surfaces were reduced and polished with no further treatment.

Group 3 (fluoride group): the fluoride gel (Nupro^®^ Acidulated Phosphate Fluoride Topical Gel (DENTSPLY Professional, 1301 Smile Way York, PA 17404, USA) was applied for on the proximal enamel surfaces for 4 min then wiped with wet gauze.

Group 4 (icon group): the Icon-Etch was applied on dry surfaces for 2 min, rinsed with water, and air-dried for a minimum of 30 s. The Icon-Dry was then applied for 30 s followed by the application of the low viscosity resin “Icon” (Icon Proximal Mini Kit, DMG, Hamburg, Germany) for 3 min and light-cured for 40 s, followed by another application for 1 min followed by light curing for 40 s.

Group 5 (MI varnish group): the MI Varnish, which is a CPP-ACP containing varnish (MI Varnish, GC, USA) was applied on clean and dry surfaces using a disposable brush.

The proximal surfaces of the teeth were cut using a low-speed precision cutter (IsoMet™, Buehler, IL, USA). The specimens were then embedded in epoxy resin.

### 2.3. Storage of the Specimens

All the specimens were stored in a remineralizing solution for 24 h, which is composed of (1.5 mM CaCl_2_, 0.9 mM NaH_2_PO_4_, 0.13 M KCl, and 5 mM NaN_3_ adjusted to pH 7.0 with HEPES.). The specimens in the MI varnish group had the MI varnish carefully removed using a surgical blade followed by gently wiping the remaining varnish with a cotton pellet soaked in diluted acetone [22]. Groups (2–5) were stored in demineralizing solution (CaCl_2_2.2 mM/L, NaH_2_PO_4_ 2.2 mM/L, acetic acid 50 mM/L, pH 4.5) for 4 days [5,6,14,15,19,20].

### 2.4. Transverse Microradiography (TMR) Assessment

All enamel blocks were embedded in epoxy resin and precisely cross-sectioned across the stripped enamel surface area with a low-speed diamond saw (Isomet 5000, Buehler, Lake Bluff, IL, USA). After this, specimens were abraded with silicon-carbide paper up to #1200 under water-cooling to a uniform thickness of 100 to 120 μm (Figure 1).

The specimens were mounted on X-ray glass plate sensitive films (High Precision Photo Plate PXHW; Konica Minolta Photo, Tokyo, Japan) together with a 15-step Aluminum step-wedge. The film, specimens, and the step-wedge were exposed to a high dosage of X-rays produced using an X-ray machine (CMR-2, SOFTEX, Kanagawa, Japan). The X-ray machine was operated at 20 kV, 2.5 mA for 10 min. The TMR glass films were exposed to developing and fixation by standardized solutions. A digital camera attached to a microscope (ML 8500, Meiji, Techno, Japan) via a specific attachment was used to obtain the images of the samples. The analysis of the results was done using two computer applications (Image J, USA) and a customized application in Microsoft Excel [5,6,14,15,19,20].

The mineral density (vol%) values were determined according to the calibration curve considering that sound enamel contains 87 vol% minerals. Values of ∆Z (Mineral density loss) and LD (lesion depth) were obtained and compared statistically. LD measurement in the current study was detected at a distance from the lesion surface where the mineral density was 5% less than that in the sound area, while ∆Z was defined by the integrated mineral loss from the surface of the lesion to the lesion depth [5,6,14,15,19,20].

### 2.5. Statistical Analysis

Descriptive statistics for lesion depth and mineral loss using mean and standard deviations were calculated. Statistical analyses for group comparison using one-way ANOVA with the Tukey correction test were used to assess the significant difference between the groups. The analyses were performed with statistical software (IBM SPSS Statistics 20.0) at a significance level of α = 0.05 [21].

## 3. Results

The IPR-reduced enamel surfaces exposed to the acidic challenge expressed significantly different patterns of acid resistance. The transverse microradiography (TMR) images for the groups are shown in (Figure 2).

The MI varnish group recorded the least mean ∆Z “Mineral density loss” (1441.27 ± 287.1) when compared to the treatment groups confirming that MI varnish exhibited the highest protective potential for IPR-prepared enamel surfaces when compared to the two tested agents (Icon 2309.43 ± 533.03, Fluoride 2405.94 ± 199.16) which did not show a significant protective effect against the adopted acidic challenge (*p* < 0.05). Moreover, the MI varnish group specimens recorded the smallest lesion depth compared to the treatment groups, confirming the protective potential of MI varnish when compared to Icon 203.7 + 30.14 µm and fluoride 226.5 + 21.76 µm (*p* < 0.05) (Table 2).

## 4. Discussion

The null hypothesis adopted in this study was partially accepted because MI Varnish was the only agent that significantly protected the interproximal reduced enamel surfaces from the simulated cariogenic acidic attack. The storage of the specimens in remineralizing solution for 24 h was carried out because previous research showed that the maximum fluoride and the inorganic components of the MI varnish are released within the first 24 h; moreover, most remineralizing varnishes stay in situ only for 24 h [23]. Previous reports showed that the acidic challenge adopted in the currently presented experiment succeeded in generating a subsurface demineralized zone of reasonable depth resembling the enamel subsurface lesions obtained using the pH-cycling procedures [5,6,14,20,24].

Our results showed no significant difference between the control and fluoride gel group in the mean mineral loss and lesion depth which agrees with multiple studies that used fluoride in its gel form and analyzed its short-term effect using TMR [5,6,14,20,25]. Previous studies suggested that the high affinity of fluoride binding to hydroxyapatite leads to its deposition and interaction with the superficial layers of enamel and hinders the diffusion of the fluoride ions within the body of the demineralized sub-surface lesions [5,6,14,20]. It is speculated that this superficial layer was easily demineralized by the 4-day demineralization challenge adopted in the current study.

A recent study utilizing the µCT showed that resin infiltration has been shown to significantly reduce mineral loss of enamel after exposure to acidic challenge [26], which contrasts our results that showed no significant effect of using the resin infiltration after IPR. The discrepancy between our results and the aforementioned results can be attributed to the difference in the acidic challenge adopted in both experiments. Moreover, some drawbacks may be associated with not using filters in the aforementioned µCT experiment [27], or mineral reference phantoms [28] to minimize the hard beaming effect which is a major problem that affects the reliability of the quantitative analysis of µCT results [15,19]. It is worth mentioning here that the μCT technique suffers from beam hardening that seriously affects the accuracy of the obtained results and requires the application of different filters during X-ray acquisition to mimic the accurate results obtained from the TMR technique. Moreover, the TMR technique is considered an accurate technique when compared to the SEM/EDS technique regarding the mineral content because of the frequent use of gold coating for the samples prior to its observation by SEM/EDS and the possible overlap of the phosphate peak with the sample’s electroconductive coating material [29]. In our experiment, the TMR technique was utilized which was extensively reported in the literature [5,6,14,15,19,20,25] suggesting that the method adopted in the currently presented study may be more reliable; however, it Is recommended to support these results with other techniques in future experiments.

The reason for obtaining no protection for the enamel after resin infiltration may be attributed to the poor penetration of the “Icon” tested resin into the enamel structure. The resin infiltration technique needs surfaces that are already demineralized and porous to allow penetration of resin particles [30,31]. In the current study, the amount of enamel reduction was approximately 300 µm which may be the same or less than the thickness of the prismless enamel which is known for its high acid resistance [9]. This high-resistant surface [9] might have limited the acid etching effect of the etchant supplied with the “Icon” system leading to diminished penetration of the “Icon” resin within the enamel structures.

Our results demonstrated that CPP-ACP containing varnish protected the stripped enamel against the acidic attack which is in agreement with previous research [32]. Treating the enamel surface with MI varnish causes the stabilization of calcium, phosphate, and hydroxide ions as amorphous nanocomplexes and localizes these ions at the tooth surface [33,34]. This may be attributed to the consistency of the MI varnish that preserved calcium and phosphate ions in their stabilized form (with the aid of CPP) without being diluted by artificial saliva [14,35]. Moreover, it may be speculated that many of the available calcium and phosphate ions were capable of diffusion through the outer surface of the enamel to form strong calcium phosphate complexes that rendered the enamel surface more resistant to the demineralization attack [14,35] adopted in the current experiment. Moreover, previous research showed that MI varnish has a good ability for penetrating and remineralizing the lesion body when compared to other remineralizing agents [14,25]. It is worth mentioning here that the percentage of CPP-ACP in the MI varnish is approximately 5% (due to the components added to its composition that render it retentive to the tooth structure.), which is relatively low when compared to 10% wht CPP-ACP content of the MI paste. However, the unique consistency of the MI varnish provides good adhesion properties to the tooth structure and renders the varnish capable of inducing a significant protective effect for proximal surfaces after the IPR procedure in contrast to the weak remineralization potential of the MI paste that was reported previously [10].

Within the limitation of this investigation, we utilized 300 µm of interproximal enamel stripping with polishing. Further studies are needed to assess the impact of polishing on the enamel structure. Moreover, a comprehensive study should be conducted to elaborate on the resistance of enamel to dissolution in relation to the enamel histological structure, and the exact role of the prism-free enamel layer in retarding the progress of acids into the enamel structure. The current study was in vitro; however, the oral environment as well as patients’ variability in oral hygiene, saliva properties, and dietary intake might give different results, as mentioned before. Thus, further in vivo and ex vivo studies are needed to determine the suitability of the current techniques and provide a better understanding of the need for protective agents after enamel stripping. The current study had a limited period for observation (approximately 5 days) and MI varnish was applied only once (it is recommended by the manufacturer to be applied every 3 months), thus increasing the observation period and testing the effectiveness of the repeated application of the MI varnish is suggested to confirm the results obtained in the current experiment.

## 5. Conclusions

Within the limitations of this in vitro study, it may be concluded that the CPP-ACP containing varnish (MI Varnish) may be considered as an agent for protecting the proximal enamel surface after interproximal reduction (IPR). New IPR guidelines should be suggested to guarantee the resistance of enamel to demineralization oral challenges after concluding the IPR procedures.

## Figures and Tables

**Figure 1 biomimetics-08-00082-f001:**
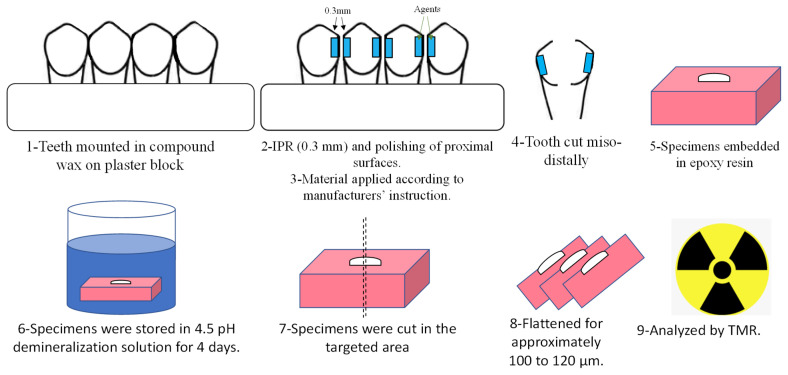
Summary of the experimental procedures.

**Figure 2 biomimetics-08-00082-f002:**
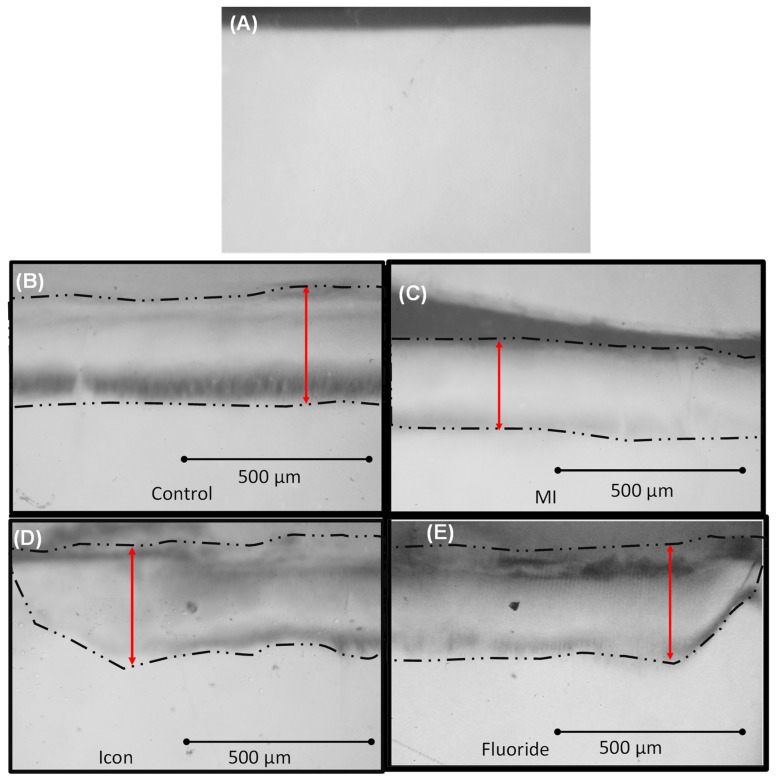
TMR images for all groups. Double arrows denote lesion depth. (**A**) Group 1: un-demineralized enamel group. (**B**) Group 2: control demineralized enamel group. (**C**) Group 3: MI varnish group. (**D**) Group 4: icon group. (**E**) Group 5: fluoride group. Double red arrows denote lesion depth.

**Table 1 biomimetics-08-00082-t001:** Materials used in this study.

Materials	Composition	Procedures
MI varnish (GC)	30–50% polybinyl acetate, 10–30% hydrogenated rosin, 20–30% ethanol, 1–8% sodium fluoride, 1–5% silicon dioxide	Applied and left undisturbed for 24 h After 24 h MI varnish was carefully removed using a surgical blade without touching surface Wiped by wet gauze Soaked in 50% acetone solution for 5 min
Icon–Etch (DMG)	15% hydrochloric acid, water, pyrogenic silica, surfactant, pigments	Apply the gel and leave it for 2 min Remove excess material with a cotton roll Rinse with water for 30 s Dry with oil—free and water—free air
Icon–Dry (DMG)	Ethanol 99%	Apply an ample amount of material and left it set for 30 s Dry with oil—free and water—free air
Icon–Infiltrant (DMG)	TEGDMA—based resin, initiators and stabilizers	Apply an ample amount of Icon—Infiltrant onto the etched surface Let Icon—Infiltrant set for 3 min Remove excess material with a cotton roll and dental floss Light—cure Icon—Infiltrant for 40 s Repeat the application and let set for 1 min, followed by curing for 40 s
Nupro^®^ Acidulated Phosphate Fluoride (DENTSPLY)	2.59% sodium fluoride (1.23% fluoride ion)	Remove excess material with a cotton roll and dental floss, and light—cure for a minimum of 40 s Apply and leave undisturbed for 4 min Wipe with wet gauze

**Table 2 biomimetics-08-00082-t002:** Mineral loss (∆ Z) and Lesion depth for all groups.

		Control Udemineralized	Control Demineralized	Icon	Fluoride	MI Varnish
∆ Z (Vol% × µm)	Mean	0	2699.47	2309.43	2405.94	1441.28 *
	Standard deviation	0	529.35	533.03	199.16	287.18
Lesion depth (µm)	Mean	0	226.04	203.73	226.47	136.97 **
	Standard deviation	0	34.62	30.14	21.76	20.06

(*) Denotes the statistically significant value of “MI varnish” ∆ Z when compared to “Control undemineralized”, “Control demineralized”, “Icon” and “Fluoride” groups (*p* < 0.05). (**) Denotes significant statistical value of “MI varnish” Lesion depth when compared to “Control undemineralized”, “Control demineralized”, “Icon” and “Fluoride” groups (*p* < 0.05).

## Data Availability

All data are available upon request.

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
