# Peer review of "Evaluation of Resin Infiltration, Fluoride and the Biomimetic Mineralization of CPP-ACP in Protecting Enamel after Orthodontic Inter-Proximal Enamel Reduction"

_biomimetics, 2023, doi:10.3390/biomimetics8010082_

Round 1

Reviewer 1 Report

Comments to the Authors

This study aimed to investigate and compare the effectiveness of different agents for protecting enamel proximal surfaces against acidic attack after interproximal reduction using Trans micro radiography technique. However, some details were missing in the methods, and some conclusions were also not reasonable. The experimental methods were too tenuous, and the evidence was insufficient.

The following issues need to be clarified:

1. “Protracting” in Line 26 may be misspelled, and protecting may be right.

2. The operation process of MI varnish group should be described more specifically. A total of 30 teeth and 60 proximal surfaces were used, but how many teeth or proximal surfaces used in each experimental group was unclear.

3. If the acidic challenge procedure was used to treat the enamel samples, the procedure of immersion in artificial saliva for 7 days should be added according to the corresponding literature cited by the author.

4. The acidic challenge procedure treatment of enamel samples is inconsistent with the actual situation of proximal enamel in the mouth, so it is recommended to replace it with the pH cycle test or replenish the pH cycle test.

5. The experimental data in present study were too simple to supply effective evidence support. It is suggested to replenish surface hardness (SH) measure, SEM, EDX or corresponding element mapping image, as well as μCT test or other relevant experiment, in order to obtain cross-validation.

5. The experimental groups lacked self-comparison before and after the acidic challenge procedure, so at least the TMR images of the experimental groups before the acid challenge experiment should be added.

6. Figure 2 was missed. In Figure 3, image A and B had similar LD length according to the red arrows and the scale, which was inconsistent with the values in Table 2. Moreover, the scale was not aligned in Figure 3.

7. The paper was not well organized and written, and the language needs to be modified by a language expert.

In conclusion, I suggest rejection.

Author Response

  1. “Protracting” in Line 26 may be misspelled and protecting may be right.

Response of the authors:

We wish to apologize for this mistake.  The English language in the manuscript was thoroughly revised.  Modified parts are in red font.

  1. The operation process of MI varnish group should be described more specifically. A total of 30 teeth and 60 proximal surfaces were used, but how many teeth or proximal surfaces used in each experimental group was unclear.

Response of the authors:

We wish to apologize for not clarifying these points.  The following parts were added to the text:

The proximal enamel surfaces were then randomly divided into 5 groups according to the protective agents applied after the IPR procedure.  Each group was comprised of 15 specimens. 

All the specimens were stored in remineralizing solution for 24 hours which is composed of (1.5mM CaCl2, 0.9mM NaH2PO4, 0.13M KCl and 5mM NaN3 adjusted to pH 7.0 with HEPES.).  The specimens in the MI varnish group had the Mi varnish carefully removed using a surgical blade followed by gently wiping the remaining varnish by a cotton pellet soaked in diluted acetone.  All specimens were stored in demineralizing solution (CaCl22.2 mM/L, NaH2PO4 2.2 mM/L, acetic acid 50 mM/L, pH 4.5) for 4 days (1-6). 

  1. If the acidic challenge procedure was used to treat the enamel samples, the procedure of immersion in artificial saliva for 7 days should be added according to the corresponding literature cited by the author.

Response of the authors:

We wish to thank the honourable reviewer for the valuable scientific comments.  The following part was added to the discussion section.

The storage of the specimens in remineralizing solution for 24 hours was carried out because previous research showed that the maximum fluoride and the inorganic components of the MI varnish are released within the first 24 hours, moreover, most remineralizing varnishes stay in situ only for 24 hours.

  1. The acidic challenge procedure treatment of enamel samples is inconsistent with the actual situation of proximal enamel in the mouth, so it is recommended to replace it with the pH cycle test or replenish the pH cycle test.

Response of the authors:

We wish to thank the honourable reviewer for the valuable scientific comments.  The following part is mentioned in the discussion section:

Previous reports showed that the acidic challenge adopted in the currently presented experiment succeeded in generating a subsurface demineralized zone of reasonable depth resembling the enamel subsurface lesions obtained using the pH-cycling procedures(1-4, 7).  

We wish to elaborate that in the current experiment we followed the technique of storing the samples in the demineralization samples continuously for 4 days without any remineralization cycles in between, this technique was previously reported in literature and may mimic the condition of high caries risk orthodontic patients who are suffering from extended demineralization cycles associated with the increased bacterial biofilm formation rate in orthodontic patients.  Previous publications adopting the storage in acidic challenge without any intermittent remineralization cycles may include:

  1. The efficacy of a bioglass (45S5) paste temporary filling used to remineralize enamel surfaces prior to bonding procedures. J Dent. 2019;85:33-8.
  2. Increasing the efficiency of CPP-ACP to remineralize enamel white spot lesions. J Dent. 2018;76:52-7.
  3. Novel evaluation and treatment techniques for white spot lesions. An in vitro study. Orthod Craniofac Res. 2017;20(3):170-6.
  4. Evaluation of Bioactive Glass and Low Viscosity Resin as Orthodontic Enamel Sealer: An In Vitro Study. J Funct Biomater. 2022;13(4).
  5. Fluoride bioactive glass paste improves bond durability and remineralizes tooth structure prior to adhesive restoration. Dent Mater. 2021;37(1):71-80.
  6. Characterization of a novel enamel sealer for bioactive remineralization of white spot lesions. J Dent. 2021;109:103663.
  7. Effects of zinc fluoride on inhibiting dentin demineralization and collagen degradation in vitro: A comparison of various topical fluoride agents. Dent Mater J. 2016 Oct 1;35(5):769-775.

h- Remineralization of enamel subsurface lesions using toothpaste containing tricalcium phosphate and fluoride: an in vitro µCT analysis.  BMC Oral Health. 2020 Oct 27;20(1):292. doi: 10.1186/s12903-020-01286-1.

  1. The experimental data in present study were too simple to supply effective evidence support. It is suggested to replenish surface hardness (SH) measure, SEM, EDX or corresponding element mapping image, as well as μCT test or other relevant experiment, in order to obtain cross-validation.

Response of the authors:

We wish to thank the honourable reviewer for the valuable scientific comments.  The following part was added to the discussion section.

It worth mentioning here that the μCT technique suffers from beam hardening that seriously affects the accuracy of the obtained results and require the application of different filters during x-ray acquisition to mimic the accurate results obtained from the TMR technique.  Moreover, the TMR technique is considered an accurate technique when compared to the SEM/EDS technique regarding the mineral content because of the frequent use of gold coating for the samples prior to its observation by SEM/EDS and the possible overlap of the phosphate peak with the gold peak(8).  Although one of the most accurate techniques utilized for the remineralization/demineralization experiments was utilized in the current experiment however, it is recommended to support these results with other techniques in future experiments.  

We wish to express our appreciation for the honourable reviewer for suggesting these techniques of reconfirmation which are not currently available for the authors however we wish to shed the light on some of the drawbacks of the suggested techniques. 

Regarding the μCT test:

We thank the honourable reviewer for the valuable scientific suggestion however we wish to quote the following statements from:

Enamel Lesion Parameter Correlations between Polychromatic Micro-CT and TMR H. Hamba , T. Nikaido , A. Sadr , S. Nakashima , and J. Tagami.

The Mineral  density (MD) assessment of teeth is important in demineralization and remineralization studies, providing an insight into the changes associated with spatial distribution of minerals within the lesions. The use of radiographic methods has been recommended to quantify MD (Ten Bosch and Angmar-Mansson, 1991). Currently, transverse microradiography (TMR) is considered as the gold standard two-dimensional (2D) technique for the determination of MD in vitro. However, TMR requires destructive sample preparation, and longitudinal analysis on the same lesion involves preparation of fragile single sections, which may be lost in the course of a study (Mellberg et al., 1986; Thomas et al., 2006). X-ray computed tomography (CT) is a non-destructive imaging method where individual projections (radiographs) are used to reconstruct a threedimensional (3D) structure. Micro-focus x-ray CT (µCT) uses a focused beam providing higher resolution on smaller samples in vitro (Elliott and Dover, 1982). The method has been frequently used in experiments exploring bone (Schweizer et al., 2007; Burghardt et al., 2008; Nazarian et al., 2008) and is a promising technique for the assessment of tooth MD (Kinney et al., 1995; Anderson et al., 1996; Davis and Wong, 1996; Lo et al., 2010; Hamba et al., 2011; Cochrane et al., 2012). The beam-hardening effect caused by the polychromatic x-ray has been shown to be a major drawback (Mulder et al., 2004; Zou et al., 2011). In short, the low-energy photons are eliminated from the beam at a faster rate than the higher-energy levels. Consequently, the beam becomes more penetrating, or harder, as it traverses through matter, creating an apparent higher attenuation near the periphery of a homogeneous sample (Jennings, 1988; Kovacs et al., 2009; Hamba et al., 2011). Several approaches that minimize inaccuracies arising from the beam-hardening effect have been recommended; in particular, beam filtrations or pre-hardening with attenuating “hardware” metal filters (aluminum, copper, or brass) (Jennings, 1988; Meganck et al., 2009) during the scanning, “software” beam-hardening correction (BHC) during reconstruction (Burghardt et al., 2008), and MD calibration have been suggested (Wong et al., 2004; Zou et al., 2011). Nevertheless, a standard metal filter for effective correction of beam-hardening on enamel has not been introduced (Zou et al., 2011). A similar trend has been observed between the linear attenuation coefficient of µCT and lesion depth (LD) of TMR in enamel subsurface lesions (Lo et al., 2010). The polychromatic µCT potentially provides a user-friendly and costeffective laboratory method for the evaluation of lesions (Zou et al., 2011); however, there appears to be no published study evaluating the correlations in

Regarding the SEM/EDS:

We thank the honourable reviewer for the valuable scientific suggestion however we wish to quote the following statements from:

SCANNING VOL. 35, 141–168 (2013)

Is Scanning Electron Microscopy/Energy Dispersive X-ray Spectrometry (SEM/EDS) Quantitative?

Summary: Scanning electron microscopy/energy dispersive X-ray spectrometry (SEM/EDS) is a widely

applied elemental microanalysis method capable of identifying and quantifying all elements in the periodic table except H, He, and Li. By following the “k-ratio” (unknown/standard) measurement protocol development for electron-excited wavelength dispersive spectrometry (WDS), SEM/EDS can achieve accuracy and precision equivalent to WDS and at substantially lower electron dose, even when severe X-ray peak overlaps occur, provided sufficient counts are recorded. Achieving this level of performance is now much more practical with the advent of the highthroughput silicon drift detector energy dispersive Xray spectrometer (SDD-EDS). However, three measurement

issues continue to diminish the impact of SEM/EDS: (1) In the qualitative analysis (i.e., element

identification) that must precede quantitative analysis, at least some current and many legacy software systems are vulnerable to occasional misidentification of major constituent peaks, with the frequency of misidentifications rising significantly for minor and trace constituents. (2) The use of standardless analysis, which is subject to much broader systematic errors, leads to quantitative results that, while useful, do not have sufficient accuracy to solve critical problems, e.g. determining the formula of a compound. (3) EDS spectrometers have such a large volume of acceptance that apparently credible spectra can be obtained from specimens with complex topography that introduce uncontrolled geometric factors that modify X-ray generation and propagation, resulting in very large systematic errors, often a factor of ten or more.

  1. The experimental groups lacked self-comparison before and after the acidic challenge procedure, so at least the TMR images of the experimental groups before the acid challenge experiment should be added.

Response of the authors:

We thank the honourable reviewer for the valuable scientific suggestion.  The TMR image for the control untreated group before acidic challenge was added.

  1. Figure 2 was missed. In Figure 3, image A and B had similar LD length according to the red arrows and the scale, which was inconsistent with the values in Table 2. Moreover, the scale was not aligned in Figure 3.

Response of the authors:

We thank the honourable reviewer for the valuable scientific suggestion.  The TMR images were replaced with other clear labelled images.

  1. The paper was not well organized and written, and the language needs to be modified by a language expert. In conclusion, I suggest rejection.

Response of the authors:

We thank the honourable reviewer for the valuable opinion.  The English language was reviewed by an English expert. 

Reviewer 2 Report

Dear Authors the paper is really interesting
First, i ask you to check the plagiarism of your article using specific sites to get a similitary report

I suggest you to modify it and add the type of article.

- The introduction section is very short and is needed to add other references to increase the quality of the manuscript

Useful paper:

[https://doi.org/10.3390/app12094646]; [ https://doi.org/10.1016/j.jebdp.2022.101786], [DOI: 10.1186/s40510-022-00438-z]

You need to review the grammar and English of your article, with the help of a native English speaker (you can specify who helped you in reviewing English in the acknowledgements) or simply by using a site that can support you in English

Please expand conclusion section with main results and future perspectives of this study

Thank You,

Kind Regards

Author Response

Comments of the honorable reviewer:

Dear Authors the paper is really interesting
First, i ask you to check the plagiarism of your article using specific sites to get a similitary report

I suggest you to modify it and add the type of article.

Response of the authors

We wish to thank the honorable reviewer for his valuable scientific comments.  The whole manuscript was revised for plagiarism. 

Comments of the honorable reviewer:

- The introduction section is very short and is needed to add other references to increase the quality of the manuscript

Useful paper:

[https://doi.org/10.3390/app12094646]; [ https://doi.org/10.1016/j.jebdp.2022.101786], [DOI: 10.1186/s40510-022-00438-z]

Response of the authors

We wish to thank the honorable reviewer for his valuable scientific comments.  The following parts were added to the manuscript. 
Previous studies showed that the first signs of enamel demineralization appear 1 month after bonding the fixed orthodontic appliances to the teeth which is manifested as white demineralized enamel spots without losing its outermost enamel surfaces[3], however these lesions will rapidly be transformed into cavitated enamel lesions if the proper oral hygiene measures will not be maintained by the patient[3].  Thus, there is a great importance in conducting the interproximal reduction (IPR) procedures in many of those cases to avoid extraction or expansion and relief the mild or even the moderate crowding, which may lead to shortening the treatment procedures, moreover the wide spread of using the Orthodontic clear align-ers increased the frequency of utilizing the IPR technique to achieve Orthodontic treatment plans goals in the shortest possible time[4]. 

Comments of the honorable reviewer:

You need to review the grammar and English of your article, with the help of a native English speaker (you can specify who helped you in reviewing English in the acknowledgements) or simply by using a site that can support you in English.

Response of the authors

We wish to thank the honorable reviewer for his valuable scientific comments.  The whole manuscript was reviewed.

Comments of the honorable reviewer:

Please expand conclusion section with main results and future perspectives of this study

Response of the authors

We wish to thank the honorable reviewer for his valuable scientific comments. The conclusion part was modified.

  1. Conclusions

Within the limitations of this invitro study it may be concluded that the CPP-ACP containing varnish (MI Varnish) may be considered as an agent for protecting the proximal enamel surface after interproximal reduction (IPR).  New IPR guidelines should be suggested to guarantee the resistance of enamel to demineralization oral challenges after concluding the IPR procedures.  

Round 2

Reviewer 1 Report

1. The experimental design divided the samples into five groups, and the total specimens written in this study were 60 in Line 109. So each group of samples should be 12, not 15.

2. Sample preparation and groups part described “Group 1 (Untreated enamel group): The enamel proximal surfaces were neither reduced nor challenged with acidic challenge.However, storage of the specimens stated that all samples were subjected to acidic challenge procedure during the experiment, which is inconsistent with the above description.

3. TMR photographs before and after the acidic challenge procedure should be provided for all experimental groups, not only for untreated control group.

4. Test data of Group 1 should be added to TABLE 2.

5. According to the literatures about acidic challenge, storing the specimens in remineralizing solution should be after the treatment that stored the specimens in demineralizing solution. This is inconsistent with the present study. Please explain why.

6. According to the literature you provided, μCT is a more promising technique for the assessment of tooth MD, and TMR requires destructive sample preparation that illustrates the limitations of TMR technology , so the data of μCT test should be added to better support your point of view. In addition, SEM-EDX test and micro-hardness test are widely recognized measurements for demineralized sample characterization and this study should supply these two tests.

In conclusion, I suggest rejection.

Author Response

Comments of the honorable reviewer:

  1. The experimental design divided the samples into five groups, and the total specimens written in this study were 60 in Line 109. So each group of samples should be 12, not 15.

Response of the authors:

We apologize for this writing mistake.  The manuscript was revised and the writing mistakes were corrected.

Comments of the honorable reviewer:

  1. Sample preparation and groups part described “Group 1 (Untreated enamel group): The enamel proximal surfaces were neither reduced nor challenged with acidic challenge.”However, storage of the specimens stated that all samples were subjected to acidic challenge procedure during the experiment, which is inconsistent with the above description.

Response of the authors:

We apologize for this writing mistake.  The following part was modified in the manuscript.

Groups (2-5) were stored in demineralizing solution (CaCl22.2 mM/L, NaH2PO4 2.2 mM/L, acetic acid 50 mM/L, pH 4.5) for 4 days [10, 11, 15-17, 20]. 

Comments of the honorable reviewer:

  1. TMR photographs before and after the acidic challenge procedure should be provided for all experimental groups, not only for untreated control group.

Response of the authors:

We thank the honorable reviewer for the valuable scientific comment.  A group for the undemineralized enamel surface was added to the experimental groups.  TMR is obtained for each specimen by sectioning, thus it is difficult to obtain the surface of enamel before/after the acid challenge.

Comments of the honorable reviewer:

  1. Test data of Group 1 should be added to TABLE 2.

Response of the authors:

We thank the honorable reviewer for the valuable scientific comment.  The data was added to the table.

Comments of the honorable reviewer:

  1. According to the literatures about acidic challenge, storing the specimens in remineralizing solution should be after the treatment that stored the specimens in demineralizing solution. This is inconsistent with the present study. Please explain why.

Response of the authors:

We thank the honorable reviewer for the valuable scientific comment.  The specimens were placed in remineralizing agent as previously reported in research reports to allow the MI varnish to release its ions to the enamel surface, which is also consistent with the instructions of the manufacturer who advice the patients by decreasing all of the acidic oral challenges in the first 24 hours.

Comments of the honorable reviewer:

  1. According to the literature you provided, μCT is a more promising technique for the assessment of tooth MD, and TMR requires destructive sample preparation that illustrates the limitations of TMR technology , so the data of μCT test should be added to better support your point of view. In addition, SEM-EDX test and micro-hardness test are widely recognized measurements for demineralized sample characterization and this study should supply these two tests

Response of the authors:

We thank the honorable reviewer for the valuable scientific comment.  The following part was added to the manuscript:

Although one of the most accurate techniques utilized for the remineralization/demineralization experiments was utilized in the current experiment however, it is recommended to support these results with other techniques in future experiments.  
